# Insights into the Adaptation to High Altitudes from Transcriptome Profiling: A Case Study of an Endangered Species, *Kingdonia uniflora*

**DOI:** 10.3390/genes14061291

**Published:** 2023-06-19

**Authors:** Man-Li Nong, Xiao-Hui Luo, Li-Xin Zhu, Ya-Nan Zhang, Xue-Yi Dun, Lei Huang

**Affiliations:** 1College of Life Sciences, Shaanxi Normal University, Xi’an 710119, China; 2Key Laboratory of Medicinal Plant Resource and Natural Pharmaceutical Chemistry of Ministry of Education, College of Life Sciences, Shaanxi Normal University, Xi’an 710119, China

**Keywords:** *Kingdonia uniflora*, adaptation, high altitude, transcriptome

## Abstract

*Kingdonia uniflora* is an endangered alpine herb that is distributed along an altitudinal gradient. The unique traits and important phylogenetic position make *K. uniflora* an ideal model for exploring how endangered plants react to altitude variation. In this study, we sampled nine individuals from three representative locations and adopted RNA-seq technology to sequence 18 tissues, aiming to uncover how *K. uniflora* responded to different altitudes at the gene expression level. We revealed that genes that responded to light stimuli and circadian rhythm genes were significantly enriched in DEGs in the leaf tissue group, while genes that were related to root development and peroxidase activity or involved in the pathways of cutin, suberin, wax biosynthesis, and monoterpenoid biosynthesis were significantly enriched in DEGs in the flower bud tissue group. All of the above genes may play an important role in the response of *K. uniflora* to various stresses, such as low temperatures and hypoxia in high-altitude environments. Furthermore, we proved that the discrepancy in gene expression patterns between leaf and flower bud tissues varied along the altitudinal gradient. Overall, our findings provide new insights into the adaptation of endangered species to high-altitude environments and further encourage parallel research to focus on the molecular mechanisms of alpine plant evolution.

## 1. Introduction

Identifying candidate genes under natural selection has been a major goal in evolutionary studies. Since natural selection helps shape adaptation when species confront new environments, we can focus on species’ adaptation to further explore the target of natural selection. High-altitude adaptation is a common phenomenon in plants; numerous endeavors have been made on the molecular mechanism of how plants adapt to high-altitude environments [1,2]. Species inhabiting high-altitude environments must face a variety of abiotic stresses, such as reduced oxygen availability, rapid fluctuations in temperature, and high ultraviolet (UV) radiation [3,4,5]. Thanks to rapidly developed high-throughput sequencing, researchers could acquire massive amounts of data on gene expressions using RNA sequencing (RNA-Seq) technology, which makes the evolutionary study of non-model species possible [6].

Although many advances have been made on the high-altitude adaptation of non-model species, the molecular mechanism by which endangered plants adapt to high mountains remains poorly understood. Endangered plants usually have a very narrow distribution and are relatively vulnerable to changes in the environment. Hence, how these endangered plants react to altitude variation is an intriguing question. The genus *Kingdonia*, belonging to the family Circaeasteraceae (Ranunculales), is a monotypic genus that is endemic to China. As the only and most endangered species in the genus, *K. uniflora* Balf. f. et W. W. Smith has been ranked as the first-class protected plant in China for a long time. Since *K. uniflora* conserves a series of ancient traits reflecting the early-diverging eudicot [7,8], it is of great importance to investigate how *K. uniflora* adapts to the environment. In fact, *K. uniflora* is an alpine herb with a narrow distribution in high mountains (2750–3900 m). Sun et al. (2020) published the draft genome sequence of *K. uniflora*, which greatly promoted its evolutionary studies [8]. Given its unique traits and important phylogenetic position, we regard *K. uniflora* as an ideal model for exploring how endangered plants react to altitude variation. 

In the present study, we carried out a comparative transcriptome study of 18 tissues from *K. uniflora*, aiming to address the following two questions: (1) whether the gene expression patterns varied along the altitudinal gradient in *K. uniflora*, and if so, how many and what kind of differentially expressed genes (DGEs); and (2) whether different tissues of *K. uniflora* presented the same gene expression pattern when adapting to high-altitude environments. Our findings will greatly enhance our understanding of the genetic basis underlying the adaptation of *K. uniflora* to high-altitude environments and also lay the foundation for the future protection of endangered alpine plants. 

## 2. Materials and Methods

### 2.1. Sample Collection

*Kingdonia uniflora* is distributed very narrowly in China, and in this study, we selected its core distribution area: Qinling mountain (Shaanxi province). To investigate how *Kingdonia uniflora* responds to different altitudes, we selected three locations ranging from 2300–3300 m at Taibai mountain (the highest peak in Qinling mountain) to sample tissues (Figure 1 shows the *Kingdonia uniflora* population at Xiabansi, Taibai mountain). At each location, we collected three individuals that were isolated from each other by at least 5 m to represent this population. This could be viewed as biological replication within groups. For each individual, we sampled two tissues for transcriptome sequencing: a leaf and a flower bud. The criterion for leaf samples is fresh, mature ones without any withering parts. The criterion for a flower bud is a 1 cm long young bud, which means it is at an early stage of flower development. To avoid unnecessary differences, we sampled all the tissues at 12:00 a.m. in August; leaf and flower buds from the same location were sampled at the same time. All fresh tissues were stored in liquid nitrogen. In all, we contained 6 groups of tissues: A, B, C (leaf), C, D, and E (flower bud). There was a total of 18 samples for transcriptome sequencing, and detailed information is shown in Table 1.

### 2.2. RNA Extraction Library Construction and Sequencing

We adopted the Trizol reagent (Thermo Fisher, Waltham, MA, USA, 15596018) to extract the total RNAs of leaf and flower bud tissues following the manufacturer’s protocol. The total RNA quantity and purity were analyzed with the Bioanalyzer 2100 and RNA 6000 Nano LabChip Kit (Agilent, Palo Alto, CA, USA, 5067-1511). High-quality RNA samples with an RIN number >7.0 were used to construct the sequencing library. We purified mRNA from total RNA (5 μg) using Dynabeads Oligo (dT) (Thermo Fisher, MA, USA) with two rounds of purification. Then, the mRNA was fragmented into short fragments using divalent cations under elevated temperature (Magnesium RNA Fragmentation Module (NEB, cat.e6150, Ipswich, MA, USA) at 94 °C for 5–7 min). All the cleaved RNA fragments were reverse-transcribed to create the cDNA by SuperScript™ II Reverse Transcriptase (Invitrogen, cat. 1896649, Carlsbad, CA, USA), which was then used to synthesize the U-labeled second-stranded DNAs with E. coli DNA polymerase I (NEB, cat.m0209, USA), RNase H (NEB, cat.m0297, USA), and dUTP Solution (Thermo Fisher, cat. R0133, USA). An A-base was added to the blunt ends of each strand, which was prepared for ligation to the indexed adapters. Each adapter contained a T-base overhang for ligating the adapter to the A-tailed fragmented DNA. Then, we ligated dual-index adapters to the fragments and performed size selection with AMPureXP beads. When the heat-labile UDG enzyme (NEB, cat.m0280, USA) treatment of the U-labeled second-stranded DNAs was performed, the ligated products were amplified with PCR under the following conditions: initial denaturation at 95 °C for 3 min; 8 cycles of denaturation at 98 °C for 15 s, annealing at 60 °C for 15 s, and extension at 72 °C for 30 s; and then final extension at 72 °C for 5 min. The average insert size for the final cDNA libraries was 300 ± 50 bp. In the end, we performed the 2 × 150 bp paired-end sequencing (PE150) on an Illumina Novaseq™ 6000 (LC-Bio Technology CO., Ltd., Hangzhou, China) following the vendor’s recommended protocol. 

### 2.3. Sequencing of All Samples and Filtering of Clean Reads

We sequenced all the above cDNA libraries, comprising 18 samples, with the Illumina Novaseq ^TM 6000^ sequencing platform. Using the Illumina paired-end RNA-seq approach, we sequenced the transcriptome of all 18 samples, generating a total of 2 million × 150 bp paired-end reads. Raw reads containing adapters or low-quality bases that will affect the following assembly and analysis were trimmed. We further filtered the reads using Cutadapt (https://cutadapt.readthedocs.io/en/stable/, version: cutadapt-1.9, accesses on 1 August 2011) [9,10]. The parameters were as follows: (1)Removing reads containing adapters;(2)Removing reads containing polyA and polyG;(3)Removing reads containing more than 5% of unknown nucleotides (N);(4)Removing low quality reads containing more than 20% of low-quality (*q*-value ≤ 20) bases. Then, sequence quality was verified using FastQC (http://www.bioinformatics.babraham.ac.uk/projects/fastqc/, 0.11.9), including the Q20, Q30, and GC-content of the clean data. After that, a total of approximately 6G bp of cleaned, paired-end reads were produced for each sample; detailed information is given in Appendix A. We submitted the raw sequence data to the NCBI Sequence Read Archive (SRA) database with accession number PRJNA971146.

### 2.4. Alignment with the Reference Genome

We aligned the reads of all samples to the *Kingdonia uniflora* reference genome (https://www.ncbi.nlm.nih.gov/genome/?term=PRJNA587615) using the HISAT2 (https://daehwankimlab.github.io/hisat2/,version:hisat2-2.2.1) package, which initially removes a portion of the reads based on quality information accompanying each read and then maps the reads to the reference genome [8,11]. HISAT2 allows multiple alignments per read (up to 20 by default) and a maximum of two mismatches when mapping the reads to the reference. HISAT2 compared the previously unmapped reads against the database of putative junctions to construct the database of potential splice junctions [11,12,13]. 

### 2.5. Quantification of Gene Abundance 

We adopted StringTie (http://ccb.jhu.edu/software/stringtie/, version:stringtie-2.1.6) to assemble the mapped reads of each sample with default parameters [14]. All transcriptomes from all samples were merged to reconstruct a comprehensive transcriptome using gffcompare software (http://ccb.jhu.edu/software/stringtie/gffcompare.shtml, version:gffcompare-0.9.8). After the final transcriptome was generated, we estimated the expression levels of all transcripts by StringTie and ballgown (http://www.bioconductor. org/packages/release/bioc/html/ballgown.html) and performed expression abundance for mRNAs by calculating the FPKM (fragment per kilobase of transcript per million mapped reads) value [15].

### 2.6. Differentially Expressed Gene (DEG) Analysis

Gene differential expression analysis was performed by DESeq2 software 3.17 [16,17] between two different groups (and by edgeR between two samples). Differentially expressed genes were screened by the following criteria: genes with a parameter of false discovery rate (FDR) below 0.05 and an absolute fold change ≥ 2. These differentially expressed genes were used for later enrichment analysis of GO functions and KEGG pathways [18,19].

### 2.7. Relationship Analysis of Samples

We used the *R* package to carry out the correlation analysis of replicas. The Pearson correlation coefficient between two replicas was calculated to evaluate repeatability between samples. The closer the correlation coefficient approaches 1, the better the repeatability between two replicas. We also performed principal component analysis (PCA) with the princomp function of *R* (http://www.r-project.org/) to reveal the relationship among the samples.

### 2.8. GO Enrichment Analysis

GO terms in the Gene Ontology database (http://www.geneontology.org/) were adopted in this study to infer the potential function of all DEGs screened between samples. We calculated gene numbers for every term. Significantly enriched GO terms in DEGs compared to the genome background were defined by the hypergeometric test [20]. The formula for calculating the *p*-value is as follows: N is the number of all genes with GO annotation; n is the number of DEGs in N; M is the number of all genes that are annotated to a certain GO term; m is the number of DEGs in M. N stands for total background gene (TB gene number); n stands for total significant gene (TS gene number); M stands for background gene (B gene number); m stands for significant gene (S gene number). In this study, GO terms meeting this condition, with *p* < 0.05, were defined as significantly enriched GO terms in DEGs. We used Metascape (http://metascape.org/) to conduct the plots related to GO analysis in this study.

### 2.9. Pathway Enrichment Analysis (KEGG)

We adopted KEGG (Kyoto Encyclopedia of Genes and Genomes) (https://www.kegg.jp/kegg/) to conduct the pathway enrichment analysis [21]. The formula for calculating the *p*-value is as follows: N is the number of all genes with the KEGG annotation, n is the number of DEGs in N, M is the number of all genes annotated to specific pathways, and m is the number of DEGs in M. Pathways meeting this condition, with *p* < 0.05, were defined as significantly enriched pathways in DEGs. N stands for total background gene (TB gene number); n stands for total significant gene (TS gene number); M stands for background gene (B gene number); m stands for significant gene (S gene number). We used OmicStudio (https://www.omicstudio.cn/index) to conduct the plots related to GO analysis in this study.

### 2.10. Gene Set Enrichment Analysis (GSEA)

Additionally, we performed gene set enrichment analysis using the software GSEA (v4.1.0) and MSigDB to identify whether a set of genes in specific GO terms or KEGG pathways shows significant differences between two groups [22]. Briefly, we input the gene expression matrix and rank genes using the Signal2Noise normalization method. We used default parameters to calculate enrichment scores and *p*-values. GO terms and KEGG pathways that meet this condition, with |NES| > 1, NOM *p*-value < 0.05, FDR *q*-value < 0.05 in a comparison, were defined as significant AS events. The classification of alternative splicing is as follows: SE: skipped exon MXE: mutually exclusive exon A5SS: alternative 5′ splice site A3SS: alternative 3′ splice site RI: retained intron.

## 3. Results

### 3.1. Transcriptome Data of 18 Samples and Mapping Information

By using the Illumina Novaseq™ 6000 sequencing platform, we gained a range of 5.61–6.99 GB of transcriptome raw data for each sample, with an average of 6.10 GB. The valid read ratio was 96.24–98.18% in all 18 samples (Appendix A), which indicated that these data were adequate for further analyses. When mapping to the reference genome, the mapped read ratio ranged from 87.94% to 96.49%, especially the unique mapped reads, which ranged from 62.20% to 86.10%. Meanwhile, we calculated the interval distribution of mapped reads and found that the majority of mapped reads showed >30 coverage. Moreover, we found that more than 85% of mapped reads lie in exon regions (Appendix A). According to the reference genome annotation, we examined the mapped reads and found that the mapped genes from the transcriptome ranged from 23,059 to 25,241 in 18 samples. After FPKM standardization, we also calculated the gene expression interval distribution of each sample; the top 1 interval was 0.3–3.57 FI and the top 2 interval was 3.57–15 FI; more than 50% of mapped genes exhibited a 0.3–15 FI value (gene expression abundance parameter) in 18 samples; detailed information is given in Appendix A. We presented all the mapped genes with their exact expression information in Appendix A. We also depicted a boxplot, a violin plot, and the gene expression density of 18 samples. Pearson correlation analysis showed that three biological replicate samples within each group presented a high correlation, while the correlations weakened among groups (Appendix A). All of the above indicated that the transcriptomes of 18 samples from six groups in this study were appropriate for later differentially expressed gene analyses.

### 3.2. Differentially Expressed Genes (DEGs) Detected from Leaf or Flower Bud Tissue at Different Altitudes

To briefly describe the sample name, leaf tissue from low, intermediate, and high altitudes is denoted as A, B, and C, respectively; flower bud tissue from low, intermediate, and high altitudes is denoted as D, E, and F, respectively. First, we carried out pairwise comparisons within the same tissue and found that differentially expressed genes (DEGs) ranged drastically from 2408 (E vs. F) to 7603 (A vs. B). As a whole, pairwise comparisons within leaf tissue preserved much more DEGs than comparisons within flower bud tissue. Notably, the most DEGs were identified between the A vs. B group (2346 m vs. 2771 m) rather than the A vs. C group (2771 m vs. 3294 m) in leaf tissue, while the most DEGs were identified between the A vs. C group (2346 m vs. 3294 m) in flower bud tissue. In flower bud tissue, we could observe that the DEGs increased with altitude. However, such a trend could not be observed in leaf tissue. We also found that DEGs were more down-regulated than up-regulated (Figure 2). For instance, in the A vs. C comparison, the ratio of down-regulated to up-regulated genes was 2444:1233. Since we selected three different altitudes in this study, the pairwise comparison was not enough to reveal the gene expression variation across all three gradients, so we examined the DEGs among the multiple comparisons A vs. B vs. C (leaf tissue) and D vs. E vs. F (flower bud tissue); the gene expression heatmap is shown in Appendix A. In total, we screened 11,860 DEGs in comparisons A vs. B vs. C and 3460 DEGs in comparisons D vs. E vs. F; details are given in Appendix A. Leaf tissue groups preserved much more DEGs than flower bud tissue groups in this study. Interestingly, down-regulated and up-regulated genes were equivalent in comparisons A vs. B vs. C, while down-regulated genes were much more common than up-regulated ones in comparisons D vs. E vs. F.

### 3.3. Differentially Expressed Genes (DEGs) Detected from Pairwise Comparison of Leaf and Flower Bud Tissue at the Same Altitude

On the other hand, we also detected DEGs between leaf and flower bud tissue at the same altitude (Figure 2). DEGs were 3431 for comparison A vs. D, 8377 for comparison B vs. E, and 6282 for comparison C vs. F. Tissues from the A and D groups, which were sampled at the lowest altitude, presented the fewest DEGs. Tissues from the B and E groups, which were sampled at middle altitude, presented the most DEGs. Notably, DEGs were still more down-regulated than up-regulated in all three comparisons here.

### 3.4. GO, KEGG, and Gsea Enrichment Analyses of DEGs

To better comprehend the biological processes of all the above DEGs, GO, KEGG, and Gsea enrichment analyses were applied in this study. In the multiple comparisons A vs. B vs. C (Figure 3), actin filament bundle assembly ranked as the top 1 in the GO enrichment analysis. Among the top 20 significantly enriched GO terms, we were especially concerned about those that were involved with high-altitude adaptation. Thus, we should pay more attention to two GO terms here: response to light stimulus (GO:0009416) and circadian rhythm—plant (GO:0007623). For the KEGG analysis, circadian rhythm—plant was among the top 20 significantly enriched pathways. Similarly, in multiple comparisons D vs. E vs. F (Figure 4), DNA-binding transcription factor activity ranked as the top 1 in GO enrichment analysis. Among the top 20 significantly enriched GO terms, regulation of root development and peroxidase activity deserved more attention. For the KEGG analysis, cutin, suberin, and wax biosynthesis and monoterpenoid biosynthesis deserved follow-up research to dissect their roles in high altitude adaptation.

Furthermore, GO, KEGG, and Gsea enrichment analyses of DEGs in pairwise comparisons were shown in Appendix A. Here, we presented those most likely related to high-altitude adaptation. In the A vs. B comparison, responses to cold, heat, and water deprivation were significantly enriched in GO terms. Responses to heat and high light intensity were significantly enriched in the Gsea analysis. In the A vs. C comparison, responses to cold, heat, and light stimuli were significantly enriched in GO terms. Responses to temperature, high light intensity, and heat acclimation were significantly enriched in the Gsea analysis. In the B vs. C comparison, responses to heat and light stimuli were significantly enriched in GO terms. Pollen tube growth, positive regulation of seed germination, and responses to blue and far-red light were significantly enriched in the Gsea analysis. Notably, circadian rhythm—plant pathway was significantly enriched in the KEGG analysis in the above three comparisons. In the D vs. E comparison, root development and aging were significantly enriched in GO terms. Cutin, suberin, and wax biosynthesis were significantly enriched in the KEGG analysis. Aging and response to nitrogen starvation were significantly enriched in the Gsea analysis. In the D vs. E comparison, root development was significantly enriched in GO terms. Circadian rhythm—plant and response to blue and far-red light were significantly enriched in the Gsea analysis. In the E vs. F comparison, cutin, suberin, and wax biosynthesis were significantly enriched in both GO and KEGG analyses. Pollen tube growth, flower development, and response to brassinosteroid were significantly enriched in the Gsea analysis. On the other hand, tissues from the same altitude provided an ideal opportunity to explore how plants adapt to the environment in different ways. We observed that cutin, suberin, and wax biosynthesis were significantly enriched in both B vs. E and C vs. F comparisons, but were not enriched in the A vs. D comparison. Notably, the A and D groups were sampled at the lowest altitude. The overall GO and KEGG enrichment results are shown in Figure 5. Given the various significantly enriched GO terms or pathways, it is impossible to investigate them all. Hence, we emphasized nine GO terms or pathways here: response to cold, response to heat, response to water deprivation, cellular response to hypoxia, response to light stimulus, response to high light intensity, circadian rhythm—plant, cutin, suberine, wax biosynthesis, and cellular response to nitrogen starvation, all of which exhibited functions concerned with high-altitude environments such as low temperature, low oxygen, and intense sunlight. Moreover, we provided a candidate gene list from the above nine GO terms or pathways (Table 2) based on statistical significance. There were two criteria for selecting candidate genes: (1) the *p*-value and *q*-value were both <0.05; (2) GO terms of candidate genes should belong to the nine terms mentioned above. Interestingly, we observed that the genes that respond to heat mainly belong to the 17.3 kDa class II heat shock protein family; the genes that respond to hypoxia mainly belong to the lignin-forming anionic peroxidase family; and the genes that respond to high light intensity mainly belong to the heat shock 70 kDa protein family. More importantly, we also found that phytochrome B, which has been proven to play a crucial role in regulating plant flowering time, was differentially expressed between samples from different altitudes.

## 4. Discussion

### 4.1. Molecular Mechanism Underlying the High-Altitude Adaptation of K. uniflora

Elucidating the molecular mechanism of how species adapt to extreme environments has been a hotspot in evolutionary biology. Compared to low-altitude areas, the high-altitude environment is a combination of various abiotic stresses, such as reduced oxygen availability, rapid fluctuations in temperature, and high ultraviolet (UV) radiation [23]. Therefore, genes responding to the above stresses would underlie the molecular mechanism of high-altitude adaptation. To date, a series of studies had reported differentially expressed genes (DEGs) and putative pathways that may be responsible for high-altitude adaptation in different organisms based on transcriptome data. For instance, glutathione metabolism, plant—pathogen interaction, and ribosome biogenesis in eukaryotes were three predicted metabolic pathways that may be associated with the high-altitude adaptation between *Notopterygium incisum* Ting ex H. T. Chang and *Notopterygium franchetii* H.Boissieu [5]. Most of the differentially accumulated metabolites (DAMs) were enriched in flavone and flavonol biosynthesis, and the most heavily enriched KEGG pathway was related to the subcategory oxidative phosphorylation in intraspecific adaptation to high altitude in *Cyclocarya paliurus* (Batal.) Iljinsk [24]. The most significantly differentially expressed top 50 genes in the high-altitude samples were derived from plants that responded to abiotic stress, such as peroxidase, superoxide dismutase protein, and the ubiquitin-conjugating enzyme, and the KEGG pathway was related to secondary metabolites, including phenylpropane and flavonoids, in intraspecific adaptation to high altitude in *Potentilla bifurca* L. [25]. Compared to the above research, our findings showed a different picture of DEGs. In this study, we revealed that genes that responded to light stimulus and circadian rhythm genes were significantly enriched in DEGs in the leaf tissue group. Given that light intensity varied considerably along the altitudinal gradient, we regarded these genes that responded to light stimuli as candidates contributing to the adaptation of *K. uniflora* to glaring light in high altitude environments. As for the circadian rhythm genes, previous studies have proved that variation in plant circadian rhythm genes is implicated in an array of plant environmental adaptations, including growth regulation, photoperiodic control of flowering, and responses to abiotic and biotic stress. Plant circadian rhythm genes can also be reset by environmental cues such as acute changes in light or temperature [26]. Therefore, the plant circadian rhythm genes may play an important role in helping *K. uniflora* adapt to abiotic stress in high-altitude environments, such as low temperatures and UV radiation. Secondly, we observed that the top significantly enriched DEGs were quite different between leaf and flower bud tissue groups. Cutin, suberine, and wax biosynthesis and monoterpenoid biosynthesis were two notable pathways that showed significantly different expression patterns along an altitudinal gradient in the flower bud tissue group. Cutin, suberine, and wax biosynthesis were shown to be related to plants’ response to high altitude [27,28]. Guo et al. (2016) reported that the variations of leaf cuticular waxes helped Compositae plants adapt to various environmental stresses and enlarge their distribution [29]. Similarly, *K. uniflora* in high-altitude environments would face more challenges, such as drought and low temperatures. The thicker the cutin, suberine and wax accumulated on the outer layer of the flower bud, the better the plant’s protection. Thus, we suggest that the genes involved in the cutin, suberine, and wax biosynthesis pathway may contribute to the adaptation of *K. uniflora* to drought and low temperature stress. As for monoterpenoid biosynthesis, monoterpenoid is regarded as an important volatile oil in plants and mostly has the functions of attracting pollination insects, preventing animals from foraging, or coordinating the relationship between plants and environment [30]. Hence, it is plausible that the monoterpenoid biosynthesis pathway was only significantly enriched in the flower bud tissue group rather than the leaf tissue group. We suggest that the reproductive organs in *K. uniflora* may accumulate some volatile oil components, such as monoterpenoid, to better respond to altitude variations.

Aside from the above GO terms or pathways discussed, we also found out about some other valuable ones, such as responses to cold, responses to heat, responses to water deprivation, cellular responses to hypoxia, responses to high light intensity, and cellular responses to nitrogen starvation. These GO terms or pathways covered nearly all the abiotic stress that plants would face in a high-altitude environment. Therefore, the DEGs revealed in this study would greatly enhance our understanding of how *K. uniflora* responded to different altitudes.

### 4.2. Response to Altitude Variation in Different Tissues of K. uniflora

Plants respond to environmental stress in a variety of ways [31,32,33]. Different tissues may not present the same expression pattern when reacting to the same stress. Likewise, leaf and flower buds have contrasting functions in plants. When it comes to high-altitude adaptation, the living conditions are relatively harsh in high-altitude habitats, and the leaf usually responds to stress related to light or oxygen. Will the intrinsic discrepancy between leaf and flower bud tissue vary across the altitudinal gradient? To answer this, we compared the leaf and flower bud tissue groups from the same location (A vs. D, B vs. E, and C vs. F). Notably, as the lowest altitude group, the A vs. D comparison conserved the fewest DEGs (See Figure 2). A plausible explanation might be that the living conditions at 2346 m altitude were not as harsh as those at 2771 m or 3294 m altitude. The gene expression patterns of leaf and flower bud tissue tend to be more similar when the environment is more suitable. The intrinsic metabolic pathway of *K. uniflora* might be more different between leaf and flower bud tissue when adapted to higher altitudes since vegetative organs are more affected by certain ecological factors such as soil type and air moisture. Additionally, DEGs from the A vs. D comparison also showed a different picture from the other two. Cutin, suberine, and wax biosynthesis was significantly enriched in both high altitude groups B vs. E and C vs. F, while they were absent in A vs. D. Thus, we suspected that when *K. uniflora* expands to higher locations, the genes related to cutin, suberine, and wax biosynthesis accumulate differently between leaf and flower bud tissue to improve its fitness under drought, salinity, or pathogen stress.

## 5. Conclusions

*K. uniflora* is an endangered alpine herb that is distributed along an altitudinal gradient. In this study, we sampled nine individuals from three representative locations and adopted RNA-seq technology to sequence 18 tissues, aiming to uncover how *K. uniflora* responded to different altitudes on the gene expression level. We revealed that genes that responded to light stimuli and circadian rhythm genes were significantly enriched in DEGs in the leaf tissue group, while genes that were related to root development and peroxidase activity or involved with the pathways of cutin, suberin, wax biosynthesis, and monoterpenoid biosynthesis were significantly enriched in DEGs in the flower bud tissue group. All of the above genes might play an important role in the response of *K. uniflora* to various stresses, such as low temperatures and hypoxia in high-altitude environments. Furthermore, we proved that the discrepancy of gene expression pattern between leaf and flower bud tissue varied along altitudinal gradient. Overall, our findings provided new insights into the adaptation of endangered species to high-altitude environments and would encourage more parallel research to focus on the molecular mechanisms of alpine plant evolution.

## Figures and Tables

**Figure 1 genes-14-01291-f001:**
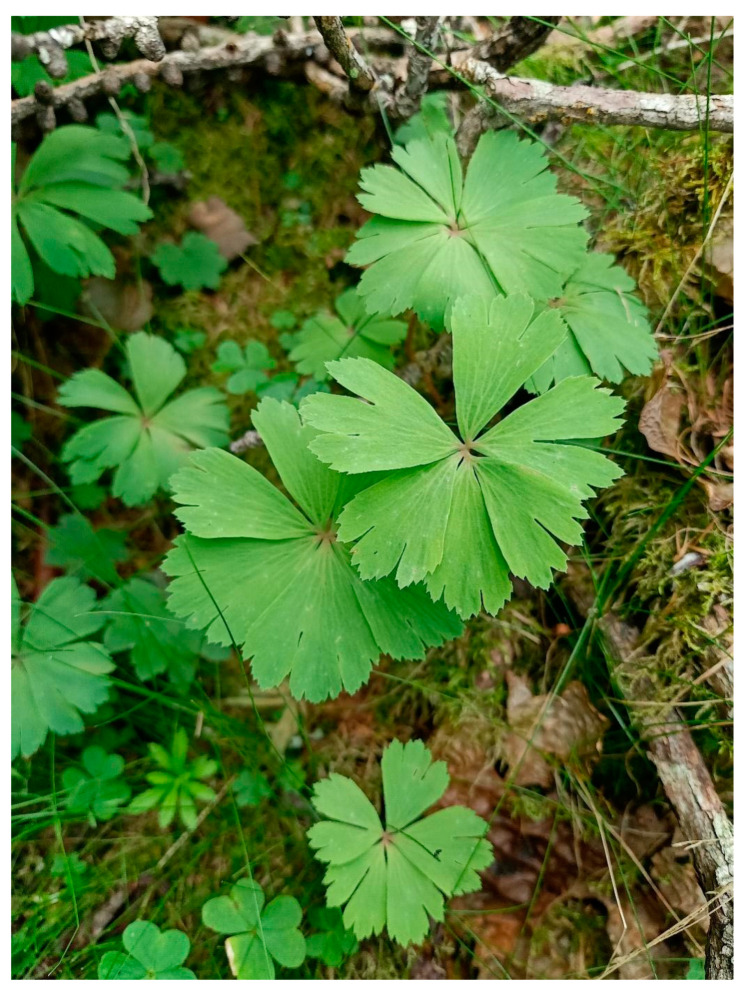
*Kingdonia uniflora* in Taibai mountain.

**Figure 2 genes-14-01291-f002:**
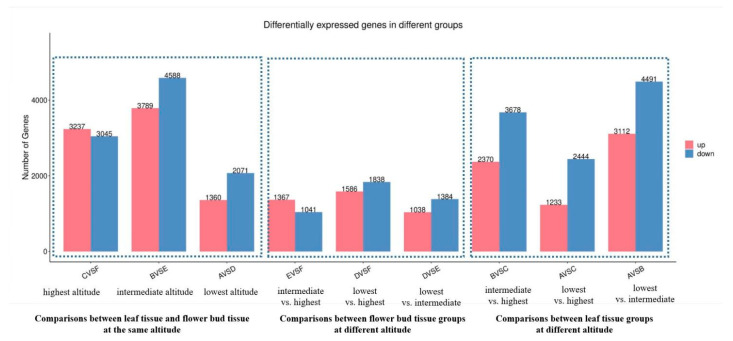
Differentially expressed genes from pairwise comparison using RNA-seq data. A, B, and C represent leaf tissue from low, intermediate, and high altitudes, respectively; D, E, and F represent flower bud tissue from low, intermediate, and high altitudes, respectively.

**Figure 3 genes-14-01291-f003:**
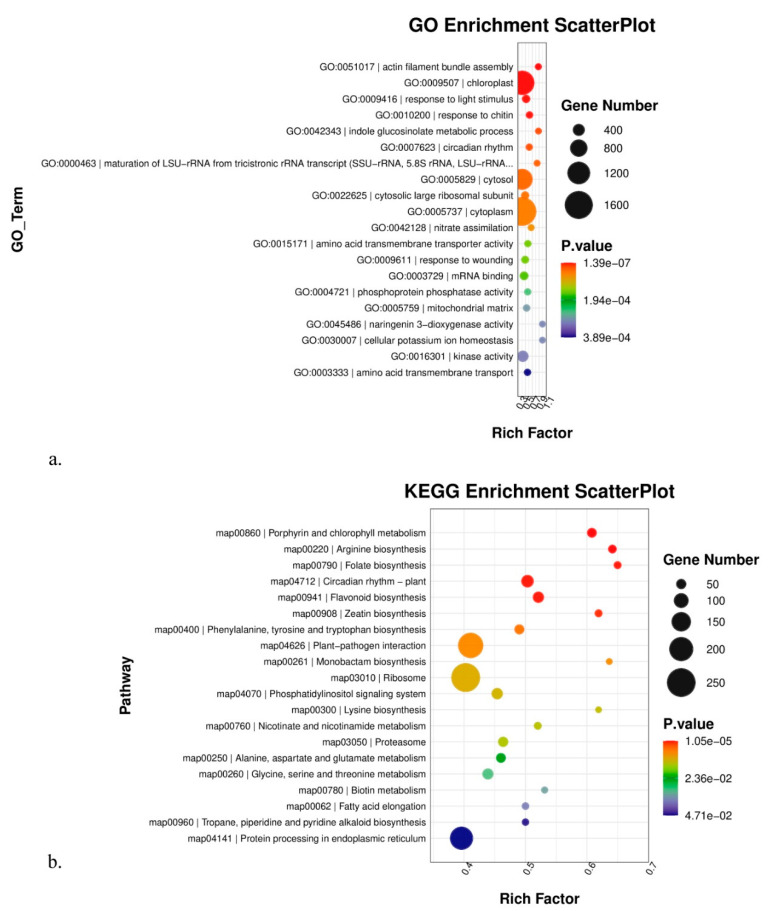
The GO enrichment scatterplot (**a**) and the KEGG enrichment scatterplot (**b**) from leaf tissue multiple comparisons A vs. B vs. C.

**Figure 4 genes-14-01291-f004:**
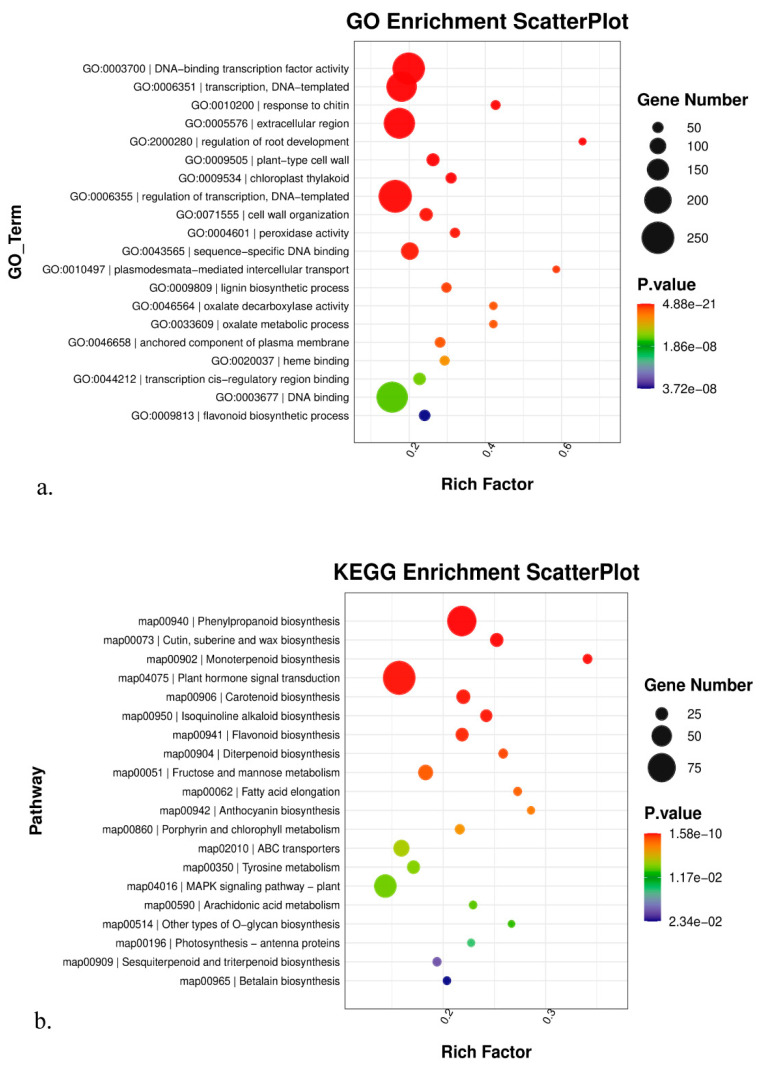
The GO enrichment scatterplot (**a**) and the KEGG enrichment scatterplot (**b**) from flower bud tissue multiple comparisons D vs. E vs. F.

**Figure 5 genes-14-01291-f005:**
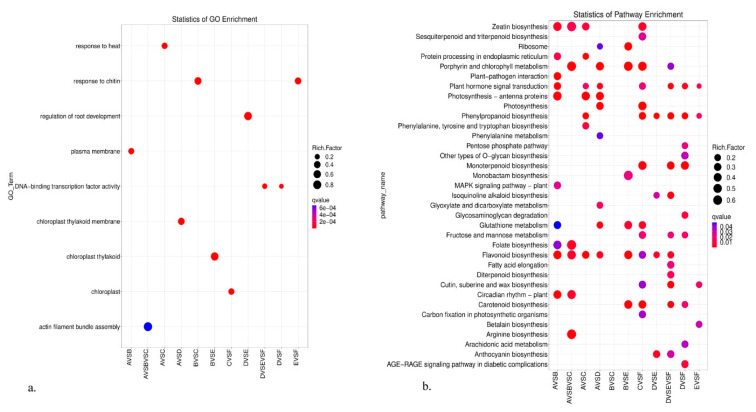
Results of the overall presentation of the GO enrichment scatterplot (**a**) and the KEGG enrichment scatterplot (**b**).

**Table 1 genes-14-01291-t001:** The geographical information and tissue type of the 18 samples used in this study.

Group Name	Location	Latitude	Longitude	Altitude (m)	Tissue Type
A	Honghegu, Meixian, Shaanxi province	34°0′46″	107°47′26″	2346	leaf
D	Honghegu, Meixian, Shaanxi province	34°0′46″	107°47′26″	2346	flower bud
B	Xiabansi, Meixian, Shaanxi province	33°42′11″	107°46′53″	2771	leaf
E	Xiabansi, Meixian, Shaanxi province	33°42′11″	107°46′53″	2771	flower bud
C	Fangyangsi, Meixian, Shaanxi province	33°58′28″	107°46′15″	3294	leaf
F	Fangyangsi, Meixian, Shaanxi province	33°58′28″	107°46′15″	3294	flower bud

**Table 2 genes-14-01291-t002:** Candidate gene list that may participate in the adaptation of *Kingdonia uniflora* to high altitude.

Gene Name	*p*-Value	*q*-Value	Putative Function
GIB67_027134	0	0	Response to heat, 17.3 kDa class II heat shock protein
GIB67_005997	0	0	Response to heat, Small heat shock protein HSP
GIB67_007978	0	0	Response to heat, 17.1 kDa class II heat shock protein-like
GIB67_028015	0	0	Response to heat, HSP20 domain-containing protein
GIB67_035570	0	0	Response to heat, 17.3 kDa class II heat shock protein
GIB67_023343	0	0	Response to cold, ACT domain-containing protein DS12, chloroplastic-like
GIB67_027084	0	0	Response to cold, cold-inducible protein
GIB67_027089	0	0	Response to cold, early light induced protein 2
GIB67_027141	0	0	Response to cold, Chlorophyll A-B binding protein
GIB67_033329	0	0	Response to cold, photosystem I chlorophyll a/b-binding protein 3-1, chloroplastic
GIB67_042678	0	0	Response to water deprivation, hypothetical protein AQUCO_00200416v1
GIB67_023028	0	0	Response to water deprivation, Stress-related protein
GIB67_035316	0	0	Response to water deprivation, plasma membrane-associated cation-binding protein 1
GIB67_037972	0	0	Response to water deprivation, hypothetical protein AQUCO_08400041v1
GIB67_017844	0	0	Cellular response to hypoxia, lignin-forming anionic peroxidase
GIB67_017843	0	0	Cellular response to hypoxia, lignin-forming anionic peroxidase
GIB67_007868	0	0	Cellular response to hypoxia, lignin-forming anionic peroxidase
GIB67_035929	0	0	Cellular response to hypoxia, lignin-forming anionic peroxidase
GIB67_035933	0	0	Cellular response to hypoxia, lignin-forming anionic peroxidase
GIB67_016048	0	0	Response to light stimulus, PREDICTED: metacaspase-4
GIB67_025148	0	0	Response to light stimulus, Chlorophyll A-B binding protein
GIB67_026789	0	0	Response to light stimulus, PREDICTED: chlorophyll a-b binding protein of LHCII type 1-like
GIB67_026779	0	0	Response to light stimulus, glyceraldehyde-3-phosphate dehydrogenase B, chloroplastic
GIB67_000124	0	0	Response to light stimulus, β tubulin1
GIB67_034645	0	0	Response to high light intensity, heat shock 70 kDa protein
GIB67_031347	0	0	Response to high light intensity, hypothetical protein AQUCO_00800081v1
GIB67_025134	0	0	Response to high light intensity, Heat shock protein 70 family
GIB67_040467	0	0	Response to high light intensity, Heat shock protein 70 family
GIB67_019867	0	0	Response to high light intensity, small heat shock protein, chloroplastic-like
GIB67_001312	0	0	Circadian rhythm—plant, phytochrome B
GIB67_007364	0	0	Circadian rhythm—plant, Cyclic dof factor 2
GIB67_001069	0	0	Circadian rhythm—plant, zinc finger protein
GIB67_035301	0	0	Circadian rhythm—plant, Chal_sti_synt_N domain-containing protein
GIB67_029338	0	0	Circadian rhythm—plant, Basic-leucine zipper domain
GIB67_008159	0.02	0.04	Cutin, suberine and wax biosynthesis, fatty acyl-CoA reductase 3-like
GIB67_019375	0.02	0.03	Cutin, suberine and wax biosynthesis, omega- hydroxypalmitate O-feruloyl transferase
GIB67_042139	0	0	Cutin, suberine and wax biosynthesis, Fatty acid hydroxylase
GIB67_038019	0	0	Cellular response to nitrogen starvation,
GIB67_011552	0	0	Cellular response to nitrogen starvation,
GIB67_026985	0	0	Cellular response to nitrogen starvation,

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
