# Peer review of "Insights into the Adaptation to High Altitudes from Transcriptome Profiling: A Case Study of an Endangered Species, Kingdonia uniflora"

_genes, 2023, doi:10.3390/genes14061291_

Round 1

Reviewer 1 Report

1. Some minor corrections have been suggested in the manuscript, which may be seen and accepted.

2. Have light intensity, oxygen concentration, and temperature been measured at different altitudes?

3. Normally, many DEGs are detected when plants are subjected to abiotic stresses. Some DEGs are commonly detected in many abiotic stresses. Hence, validation is required to confirm candidate genes associated with abiotic stresses. Whether there is any planning for the validation of some critical DEGs responsible for the adaptation of  Kingdonia uniflora to high- altitude?

Author Response

Referee 1:

  1. Some minor corrections have been suggested in the manuscript, which may be seen and accepted.

We have made the corrections according to the suggestions in this version.

  1. Have light intensity, oxygen concentration, and temperature been measured at different altitudes?

Thanks for the valuable advices. We plan to measure all the above ecological factors in the follow-up studies, we just focus on the gene expression pattern across the altitude variation in Kingdonia uniflora in this study.

  1. Normally, many DEGs are detected when plants are subjected to abiotic stresses. Some DEGs are commonly detected in many abiotic stresses. Hence, validation is required to confirm candidate genes associated with abiotic stresses. Whether there is any planning for the validation of some critical DEGs responsible for the adaptation of Kingdonia uniflora to high- altitude?

Yes, we have a detailed plan to validate the potential functions of 10-15 candidate genes from the candidate gene list provided in our manuscript. We considered gene silencing and gene editing approaches to uncover how these candidate genes help Kingdonia uniflora adaptation to high-altitude. However, these works could be very time-consuming and we may not be able to present these results in this manuscript.

Reviewer 2 Report

In the manuscript “Insights on the adaptation to high altitude from transcriptome profiling: A case study of an endangered species Kingdonia uniflora" Nong et al. reported the thoroughly how K. uniflora responded to different altitude on gene expression level. They identified light responsive and circadian rhythm regulated genes were significantly affected in leaf tissues, while root development and peroxidase activity genes, or genes involved biosynthesis of cutin, suberin, wax and monoterpenoid were differently expressed in flower bud tissue at different altitudes. The genesis of this paper is to provide new insights into the adaptation of endangered species to high altitude environments. The title and abstract are appropriate for the content of the text. The experiments were well conducted. The authors should revise the language and logic of manuscript to improve readability.

As explained below, the concerns regarding the experimental design need to be addressed in order for the main conclusions to be well-founded.

1.     In the materials and methods part, please add detailed information of sample collection time of the day.

2.     Please relocate figure 1, it should not be cited in the introduction.

3.     In line213-227, It is not easy to interpret the text, please replace what A,B,C,D,E,F stands for with sample name.

4.     Fig.2, please rewrite result 2 supported by figure 2. Is it more reasonable to compare the transcriptome in the same organ at different altitudes and divide the bar chart into several panels like comparing Leaf samples, floral bud at different altitude, and different organs at the same altitude.

5.     Figure 2 please briefly explain why less DEGs detected at lower altitude in leaves and floral buds.

6.     Figure 2, why more DEGs detected in leaves at A vs. B compared to A vs. C? Please also explain why the more dramatic altitude differences (A vs C) has less effect transcriptome?

7.     Fig.3a, please reformat the figure as fig.3b, as it is not easy to interpret.

8.     Table 2, please briefly show how these potential candidate genes were selected. In the table, please remove the starting and ending positions, and add the fold changes and P-value of each gene, also, the group of DEGs detected. This information should be more interesting to the scientific community and readers.

9.     Please verify some of these DEGs by qRT-PCR.

10.  In general, for all the figures, please provide brief information of how each figure is generated.

Author Response

Referee 2:

In the manuscript “Insights on the adaptation to high altitude from transcriptome profiling: A case study of an endangered species Kingdonia uniflora" Nong et al. reported the thoroughly how K. uniflora responded to different altitude on gene expression level. They identified light responsive and circadian rhythm regulated genes were significantly affected in leaf tissues, while root development and peroxidase activity genes, or genes involved biosynthesis of cutin, suberin, wax and monoterpenoid were differently expressed in flower bud tissue at different altitudes. The genesis of this paper is to provide new insights into the adaptation of endangered species to high altitude environments. The title and abstract are appropriate for the content of the text. The experiments were well conducted. The authors should revise the language and logic of manuscript to improve readability.

As explained below, the concerns regarding the experimental design need to be addressed in order for the main conclusions to be well-founded.

  1. In the materials and methods part, please add detailed information of sample collection time of the day. 

We have added the collection time in this version, see line99.

  1. Please relocate figure 1, it should not be cited in the introduction.

We have relocated the Fig.1, see line91-92.

  1. In line213-227, It is not easy to interpret the text, please replace what A,B,C,D,E,F stands for with sample name.

    We have added the brief introduction for A-F, see line 255-257.

  1. 2, please rewrite result 2 supported by figure 2. Is it more reasonable to compare the transcriptome in the same organ at different altitudes and divide the bar chart into several panels like comparing Leaf samples, floral bud at different altitude, and different organs at the same altitude.

   We have made some corrections in this section, see line261-266.

  1. Figure 2 please briefly explain why less DEGs detected at lower altitude in leaves and floral buds.

We have added the plausible explanations, see line423-429.

  1. Figure 2, why more DEGs detected in leaves at A vs. B compared to A vs. C? Please also explain why the more dramatic altitude differences (A vs C) has less effect transcriptome?

Gene expression pattern varied very sensitively to environment changes, thus dramatic altitude differences between A and C did not guarantee all the ecological factors varied a lot between A and C. In the same way, the ecological factors between A and B could be very different even though their altitude were not so different as A vs. C.

  1. Fig.3a, please reformat the figure as fig.3b, as it is not easy to interpret.

We have made improvments in this version.

  1. Table 2, please briefly show how these potential candidate genes were selected. In the table, please remove the starting and ending positions, and add the fold changes and P-value of each gene, also, the group of DEGs detected. This information should be more interesting to the scientific community and readers.

We have added the selection criterion, see line337-340. we also added the p-value information in Table 2.

  1. Please verify some of these DEGs by qRT-PCR.

   We have checked other research articles and found that verifying some of the candidate gene is not compulsory now. Thus we do not verify them by qRT-PCR.

  1. In general, for all the figures, please provide brief information of how each figure is generated.

We provided detailed procedures of data analysis in method part, see line 204 and 216, readers may refer to those information.
